# De Novo Asp219Val Mutation in Cardiac Tropomyosin Associated with Hypertrophic Cardiomyopathy

**DOI:** 10.3390/ijms24010018

**Published:** 2022-12-20

**Authors:** Andrey K. Tsaturyan, Elena V. Zaklyazminskaya, Margarita E. Polyak, Galina V. Kopylova, Daniil V. Shchepkin, Anastasia M. Kochurova, Anastasiia D. Gonchar, Sergey Y. Kleymenov, Natalia A. Koubasova, Sergey Y. Bershitsky, Alexander M. Matyushenko, Dmitrii I. Levitsky

**Affiliations:** 1Institute of Mechanics, Moscow State University, Moscow 119192, Russia; 2Petrovsky National Research Centre of Surgery, Moscow 119991, Russia; 3Institute of Immunology and Physiology, Ural Branch of Russian Academy of Sciences, Yekaterinburg 620049, Russia; 4Research Center of Biotechnology, A.N. Bach Institute of Biochemistry, Russian Academy of Sciences, Moscow 119071, Russia; 5Koltzov Institute of Developmental Biology, Russian Academy of Sciences, Moscow 119334, Russia

**Keywords:** tropomyosin, cardiomyopathic mutations, hypertrophic cardiomyopathy, differential scanning calorimetry, in vitro motility assay, molecular dynamics

## Abstract

Hypertrophic cardiomyopathy (HCM), caused by mutations in thin filament proteins, manifests as moderate cardiac hypertrophy and is associated with sudden cardiac death (SCD). We identified a new de novo variant, c.656A>T (p.D219V), in the *TPM1* gene encoding cardiac tropomyosin 1.1 (Tpm) in a young SCD victim with post-mortem-diagnosed HCM. We produced recombinant D219V Tpm1.1 and studied its structural and functional properties using various biochemical and biophysical methods. The D219V mutation did not affect the Tpm affinity for F-actin but increased the thermal stability of the Tpm molecule and Tpm-F-actin complex. The D219V mutation significantly increased the Ca^2+^ sensitivity of the sliding velocity of thin filaments over cardiac myosin in an in vitro motility assay and impaired the inhibition of the filament sliding at low Ca^2+^ concentration. The molecular dynamics (MD) simulation provided insight into a possible molecular mechanism of the effect of the mutation that is most likely a cause of the weakening of the Tpm interaction with actin in the "closed" state and so makes it an easier transition to the “open” state. The changes in the Ca^2+^ regulation of the actin-myosin interaction characteristic of genetic HCM suggest that the mutation is likely pathogenic.

## 1. Introduction

Hypertrophic cardiomyopathy (HCM) is the most common inherited heart disease, caused by parental or de novo mutations in more than 20 genes [1]. Alterations in the genes encoding regulatory proteins of the thin (actin) filament, troponin (Tn), and tropomyosin (Tpm) are relatively uncommon and account for 5–10% of HCM cases [2,3]. The clinical phenotype of HCM caused by thin filament mutations usually shows milder left ventricular hypertrophy that nevertheless is associated with increased morbidity related to heart failure and a high risk of life-threatening arrhythmias [4,5].

Findings from a large number of studies have established that HCM is often accompanied by increased Ca^2+^-sensitivity, i.e., a shift of the force-calcium relation towards lower Ca^2+^ concentrations and residual tension even at very low Ca^2+^ concentrations [6,7,8,9,10,11,12,13,14,15]. Tropomyosin (Tpm) encoded by the *TPM1* gene is a major participant in calcium regulation of the contraction of striated muscles, including the myocardium, serving as a gatekeeper for the actin-myosin interaction [16,17,18,19]. Tpm has a coiled-coil structure and polymerizes via the head-to-tail junctions to form continuous strands on opposite sides of the entire actin filament. Tpm forms complexes with another regulatory protein, troponin (Tn) in a 1:1 stoichiometry. In the absence of Ca^2+^, Tn anchors on actin and keeps the Tpm-Tn strand in the “blocked” position, which prevents myosin from binding to actin [20]. When Tn binds Ca^2+^, it releases actin and allows the Tpm strand to move azimuthally to the “closed” position, where myosin heads can bind actin. Upon the formation of a strong actomyosin bond, myosin pushes the strand further away to the “open” position, where neighboring actin monomers also become open for myosin binding [20].

There are dozens of rare genetic variants in the *TPM1* gene registered in the ClinVar database (https://www.ncbi.nlm.nih.gov/clinvar/), which were found in genetic research studies or during clinical testing. Only a small fraction of them are followed up with sufficient clinical data and family history, and even fewer are well characterized in experimental settings. Therefore, most of these single submissions are qualified as variants of uncertain significance (Class III according to ACMG 2015 criteria [21]). Class III variants are clinically non-actionable and represent a substantial difficulty for genetic counseling in burdened families. A possible approach to solving this uncertainty in the causative role of the genetic finding is studying the structural and functional impact of the mutations using various in vitro techniques.

We identified a new genetic variant, c.656A>T (p.D219V), in the *TPM1* gene in a young sudden cardiac death victim. To uncover molecular mechanisms of Tpm-mediated cardiomyopathy, we studied the structural and functional properties of Tpm with D219V substitution, its interactions with partner proteins, and its influence on Ca^2+^ regulation of the interaction of ventricular and atrial myosin with actin. To understand the molecular mechanisms of the changes caused by the mutation, we performed molecular dynamics (MD) simulation of the structure of the regulatory unit of the thin filament formed by filamentous actin and two Tpm-Tn strands on both sides of the actin helix.

## 2. Results

### 2.1. Genetic Evaluation and Mutation Detection

Genetic testing was a part of the post mortem investigation of the sudden cardiac death (SCD) victim. A young girl, 11 years old, died suddenly without any specific trigger factor during her usual daily activities. A detailed investigation was performed following current recommendations [22]. Asymmetric non-obstructive hypertrophy of the interventricular septum (IVS) was the only finding during the autopsy. In her past medical history, she had a normal health status and tolerance to physical training with no complaints, no syncope, and no medication prescribed. Her family history was unremarkable. Genetic testing revealed a new genetic variant, c.656A>T (p.D219V), in the *TPM1* gene (Figure 1). This variant is absent in the gnomAD database [23] (last accessed 02-11-2022). In silico pathogenicity score predictions integrated by Varsome resource [24] were contradictory (11 “pathogenic” predictions vs. 8 “uncertain” predictions). Cascade familial screening has shown that neither of the parents carries this variant (Figure 1). It testifies to the de novo origin of the missense variant c.656A>T (p.D219V) in the *TPM1* gene. We classify this variant as likely pathogenic (Class IV) according to ACMG (2015) criteria [21].

To evaluate the pathogenicity of this mutation in the *TPM1* gene, we produced recombinant D219V Tpm and applied various methods to investigate how the D219V substitution affects the structural and functional properties of Tpm.

### 2.2. Effect of the D219V Mutation on Tpm Thermal Stability

The effect of the D219V mutation on the thermal stability of Tpm was studied with differential scanning calorimetry (DSC). The temperature dependence of the heat capacities of D219V and WT Tpm is shown in Figure 2.

The mutation caused a shift in the DSC plot (Figure 2) toward a higher temperature. The most obvious change was a 2.5 °C shift in the peak position of calorimetric domain 2 that was not accompanied by a significant change in the enthalpy of thermal melting (Table 1). As this domain containing the D219 residue was shown to represent the thermal unfolding of the C-terminal part of Tpm [26], the data suggest that the D219V mutation causes a local Tpm stabilization. Another significant change in the DSC data induced by the D219V Tpm mutation was a 0.9 °C shift in the peak position of the calorimetric domain 3 and a decrease in the enthalpy associated with this peak. Since this domain was shown to correspond to the N-terminal part of the Tpm molecule [26], a decrease in the enthalpy associated with this domain suggests that a part of the coiled-coil structure of this domain becomes more fusible. This suggestion is corroborated by a significant increase in the enthalpy of the least thermostable calorimetric domain 1 (Table 1).

### 2.3. Effect of the D219V Mutation on Tpm Solution Viscosity

The D219V Tpm viscosity, i.e., the viscosity of the D219V Tpm solution after the buffer viscosity subtraction, was significantly higher than that of the WT Tpm: 0.424 ± 0.002 mPa∙s vs. 0.250 ± 0.005 mPa∙s (Mean ± SEM), respectively. The increase in the viscosity might occur due to a rise in the Tpm rigidity or, more probably, to an increased formation of the head-to-tail complexes of the Tpm molecules, as the increase in the viscosity was almost 2-fold.

### 2.4. Effect of the D219V Mutation on Tpm Binding to F-Actin

The effect of the D219V mutation on the Tpm affinity to F-actin at room temperature was assessed using a co-sedimentation assay (Figure 3A). The thermal stability of the F-actin–Tpm complex was estimated with light scattering, whose thermally induced decrease reflects the dissociation of the complex (Figure 3B).

The D219V Tpm mutation statistically insignificantly increased the Tpm affinity to F-actin (Figure 3A). The *K*_50%_ values, corresponding to the Tpm concentration at which half of the actin becomes saturated, were 2.64 ± 0.1 µM for D219V Tpm vs. 2.88 ± 0.12 µM for WT Tpm. The D219V Tpm mutation significantly increased the thermal stability of the Tpm-actin complex (Figure 3B); the T_diss_ values (i.e., the temperature at which a 50% decrease in the light scattering occurs) were equal to 45.27 ± 0.03 °C for D219V Tpm vs. 43.28 ± 0.03 °C for WT Tpm. These findings correlate with an increase in the viscosity of the Tpm solution caused by the D219V mutation, as well as with the results of previous studies [27,28] indicating that the stability of the Tpm–F-actin complexes depends rather on the thermal stability of the Tpm molecule than on the Tpm affinity to actin.

### 2.5. Effect of the D219V Tpm Mutation on the Actin-Myosin Interaction and Its Ca^2+^ Regulation In Vitro

To characterize the effect of the D219V mutation on the actin-myosin interaction, we analyzed, using an in vitro motility assay, the dependence of the sliding velocity of the F-actin–Tpm filament on the surface density of myosin, which was varied by changing the myosin concentration in the flow cell (Figure 4, Table 2). Earlier, it was shown that the myosin concentration needed for bare F-actin to achieve the half-maximal velocity is significantly lower than that for the F-actin–Tpm filaments [29,30]. Here, the addition of WT Tpm decreased the F-actin velocity significantly, whereas the D219V Tpm did not affect the velocity. In addition, the F-actin–Tpm filaments with the D219V mutation required substantially less myosin concentration to achieve the half-maximal velocity than those with WT Tpm. Thus, the D219V Tpm mutation significantly affects the tropomyosin regulation of actin-myosin interaction.

To study the effects of the D219V Tpm mutation on the calcium regulation of actin-myosin interaction, we analyzed the Ca^2+^ dependence of the sliding velocity of regulated thin filaments reconstructed from F-actin, Tpm, and Tn, over myosin from the left ventricle (LV) and left atrium (LA) in the in vitro motility assay (Figure 5, Table 3).

The changes in the parameters of Ca^2+^-dependent movements of thin filaments over ventricular and atrial myosin caused by the D219V Tpm mutation were similar. The main features were as follows (Figure 5, Table 3): (i) incomplete inhibiting of filament sliding at low Ca^2+^ concentrations (*p*Ca ≥ 7); (ii) increased Ca^2+^ sensitivity characterized by Ca^2+^ concentration required for half-activation of the sliding velocity; and (iii) increased sliding velocity at saturating Ca^2+^ concentrations for both ventricular and atrial myosin. The first two features, increased Ca^2+^ sensitivity and incomplete relaxation at low Ca^2+^ concentrations, are characteristic of HCM [6,7,14,15].

Another essential characteristic of the regulation of the actin-myosin interaction is the myosin-myosin cooperativity along the thin filament [29]. For assessment of the effect of the D219V Tpm mutation on cooperativity, the dependence of the sliding velocity in vitro on myosin concentration was measured (Figure 6). A decrease in the myosin concentration reduced the myosin surface density and consequently increased the distance between myosin molecules, pulling a thin filament in vitro. The higher the cooperativity, the fewer myosin heads are needed to move the thin filament to achieve the half-maximal velocity. No effect of the D219V Tpm mutation on the myosin-myosin cooperativity was found (Figure 6).

### 2.6. Molecular Dynamics Simulation

The in vitro motility assay data suggest that the D219V Tpm mutation impairs muscle relaxation and facilitates the transition of the thin filament from the blocked to the “open” state (Figure 5, Table 3). To understand the possible molecular mechanism of these changes, we ran 200 ns-long MD simulations of the regulatory unit of the thin filament containing WT Tpm or D219V Tpm. The structure of the regulatory unit of the thin filament in the absence of calcium (Protein Data Bank PDB ID 6KN7) [31] was used as a starting model. The side chain of the Tpm D219 residue can form a hydrogen bond (*h*-bond) with the K236 residue of actin in the 6KN7 structure (Figure 7). Neighbor E218 Tpm residue and K328 actin residue are also close enough to participate in the *h*-bonding between Tpm and actin.

The time course of the *h*-bonds existence between the D219 and E218 Tpm residues and the K326 and K328 actin residues during MD trajectories is shown in Figure 8. In the MD model with WT Tpm, there was a dynamic competition between two negatively charged Tpm residues, D219 and E218, for *h*-bonding to two positively charged actin residues. The D219 Tpm residue forms an *h*-bond with the actin K326 residue for only 12-24% of the total duration of the MD trajectory (the two figures correspond to two different Tpm strands). However, with WT Tpm, an *h*-bond exists between the two pairs of residues for 45-87% of the simulation time. When the D219V mutation was introduced into the Tpm model, the 219 Tpm residue did not form any *h*-bonds with actin, and the relative total lifetime of an *h*-bond between the pairs of residues decreased by 16-56% for two Tpm-Tn strands. Thus, for about half of the MD trajectory, there were no *h*-bonds between the Tpm strand and the actin monomer located next to the 218 and 219 Tpm residues. The data suggest that the D219 Tpm mutation causes a local destabilization of the blocked state of the thin filament in the absence of Ca^2+^: the possibility of myosin binding to a neighboring actin monomer becomes higher than for WT Tpm.

## 3. Discussion

### 3.1. The D219V Tpm Mutation Increases Ca^2+^ Sensitivity and Impairs Relaxation of the Regulated Thin Filaments

The main result of our work is finding an increase in the Ca^2+^ sensitivity of the sliding velocity of reconstructed thin filaments containing D219V Tpm compared to those with WT Tpm (Figure 5, Table 3). The shift in the Ca^2+^-velocity curve by 0.3–0.4 *p*Ca units (Table 3) means that the Ca^2+^ concentration required for moving the filaments with a half-maximum velocity decreases by a factor of 2–3 upon the D219V substitution in Tpm. The residual sliding velocity of 18–19% of its maximum value at a low Ca^2+^ concentration (Figure 5, Table 3) suggests that a substantial fraction of myosin heads can bind actin in diastole, hampering the refilling of the heart chambers. Importantly, these data were obtained with cardiac muscle myosin from both the ventricle and atrium. The fact that the violation of the actin-myosin interaction with the D219V substitution in Tpm occurs not only in the ventricular myocardium but also in the atria, disrupting their function, aggravates the development of cardiac pathology in general [32,33,34]. The increase in Ca^2+^ sensitivity and incomplete relaxation are typical for HCM [6,7,10,14]. Thus, the results of the in vitro motility experiments suggest that the D219V Tpm mutation is pathogenic.

### 3.2. Possible Molecular Mechanisms of the Effects of the D219V Tpm Mutation

The only *TPM1* mutation in the ClinVar database that involves the 219 Tpm residue is the D219N one, classified as HCM-associated and likely pathogenic [4]. The results of the DSC study of D219N Tpm reported briefly in a meeting abstract [35] are similar to those found here for the D219V Tpm (Figure 2, Table 1), despite the difference in the substituting amino acids.

The molecular mechanism of the effects of the D219V Tpm mutation can be hypothesized based on previously published data. The D219 residue and its neighbor, the E218 residue, are believed to participate in the interaction with the K236 and K238 residues of the nearest actin monomer [36,37]. In recent cryo-EM structures of the regulatory units of the thin filament in the blocked and closed states, these pairs of oppositely charged residues are close enough to interact with each other [31,38].

Our MD simulation of the regulatory unit of the cardiac thin filament in the blocked state sheds some light on the possible molecular mechanism of the effect of the D219V Tpm mutation. An incomplete block of the actin-myosin interaction and its higher Ca^2+^ sensitivity can result from a decrease in the time-averaged number of *h*-bonds between E218 and D219 Tpm residues and K326 and K328 residues of actin (Figure 7 and Figure 8). Although during the MD simulation, the lifetime of an *h*-bond between the D219 residue of WT Tpm and actin was low, the substitution of negatively charged Asp with Val possibly weakened an electrostatic attraction between the negatively charged cluster of Tpm residues E218 and D219 and the positively charged cluster of actin residues K326 and K328. As a result of the weakening, the lifetime of the *h*-bonds of neighbors E218 and actin was also reduced upon the D219V mutation (Figure 8). The difference in the number of *h*-bonds between the two Tpm strands (Figure 8) was probably caused by an asymmetry of the 6KN7 structure [31]. Additionally, it is worth mentioning that the averaging over the MD trajectory number of the *h*-bonds between Tpm and the K236 and K328 actin residues of the whole structure did not change statistically significantly upon the D219V mutation. The local destabilization of the blocked state may promote the binding of a myosin head to an actin monomer close to the 219 Tpm residue in the absence of Ca^2+^. The subsequent transition of the actin-myosin bond to a strongly bound state would promote myosin binding to neighboring actin monomers. 

The results of our MD simulation of the Tpm-Tn-F-actin complex in the blocked state do not contradict the results of the in vitro motility experiments at a saturating Ca^2+^ concentration (Figure 6). The D219 Tpm mutation caused a local destabilization of the blocked state due to a reduction of the Tpm *h*-bonding to an actin monomer (Figure 8). This local change does not necessarily lead to a global shift in the myosin-based cooperativity of the close-to-open transition tested by the experiments shown in Figure 6. Recently, Baldo et al. [39] estimated the free energy required for myosin binding to actin monomers in the vicinity of different parts of Tpm, including the region between Tpm residues 180−220. The energy is not too high, so its decrease upon reduction of the *h*-bond occupancy between actin and Tpm residues 218 and 219 caused by the D219V substitution can promote myosin binding to the actin monomer controlled by this Tpm pseudo repeat even in the absence of Ca^2+^.

### 3.3. Changes in Structure-Function Relationship Caused by the D219V Tpm Mutation

An increase in the thermal stability of D219V Tpm (Figure 2, Table 1) and its complex with F-actin (Figure 3B) compared to WT Tpm shows that the mutation stabilizes the Tpm molecule, especially its C-terminal part. The change in the thermal stability of the complex was not surprising, as the thermal stability of the F-actin-Tpm complex mainly depends on the thermal stability of the Tpm molecule [28,29]. We have performed the MD simulation only at 300 K (27 °C), not at a higher temperature, so our MD modeling does not provide any information on how the *h*-bonding between Tpm and actin might change upon heating.

It is worth mentioning that we tested the effect of the D219V mutation on the thermal stability of the Tpm-F-actin complex in the absence of Tn, as the heating of the Tpm-Tn-F-actin complex causes Tn denaturation. The denaturation is accompanied by aggregation that impairs the results of the light-scattering experiments. Nevertheless, an increase in the thermal stability of the Tpm molecules caused by the D219V mutation did not affect the Tpm affinity for F-actin (Figure 3A). This fact might be explained by a combination of two opposite effects. First, an enhancement in the Tpm-Tpm binding at the overlap junctions increases the viscosity of the Tpm solution and promotes the Tpm’s binding to F-actin. Second, a local weakening in the actin-Tpm interaction *via* a local decrease in *h*-bonding, discussed above, can weaken the binding. A mechanism of strengthening the Tpm-Tpm interaction due to a mutation of a distant 219 Tpm residue is unknown. We are aware that the explanation given above is mainly hypothetical, and further studies are needed to understand the reason for the increase in the viscosity of the Tpm solution upon the D219V substitution and its relation to the Tpm affinity for F-actin.

A striking finding of our work is illustrated in Figure 4 and Table 2. While WT Tpm decreased the maximal sliding velocity in vitro, in the presence of D219V Tpm, myosin moved the F-actin–Tpm filaments at the same velocity as F-actin. In addition, D219V Tpm decreased the myosin concentration required for the filaments’ movement at a half-maximal velocity by 30–50% (Figure 4, Table 2). At the same time, co-sedimentation data showed that the affinity of WT Tpm and D219V Tpm to F-actin was identical (Figure 3A). These two results, which seem contradictory, can be reconciled by the assumption that in the absence of troponin D219V Tpm, in contrast to WT Tpm, Tpm is easily pushed into the "open" state by myosin. Such an assumption agrees well with the results of the MD simulation, which showed that the mutation weakens the interaction of tropomyosin with actin (Figure 8). 

In completely reconstructed thin filaments, the difference in the velocity was not so great. In the presence of troponin, a 12–15% velocity excess of thin filaments with D219V Tpm over those with WT Tpm at a saturated calcium concentration (Figure 6) can also be explained by an easier transition of Tpm with the mutation to the "open" state. However, in this case, such a transition can be partially restricted by troponin. The D219V mutation did not influence the myosin half-maximum concentration of the filament’s velocity, which is probably affected by the Tpm-Tn interaction.

### 3.4. Relation to Previous Works

Effects similar to those reported here were found for the E218L substitution of the neighboring Tpm residue [40]. An increase in the Ca^2+^ sensitivity of the sliding velocity of the thin filaments in vitro with E218L Tpm, as well as an increase in maximum sliding velocity and residual Ca^2+^-free filament motility compared to WT Tpm. Again, no changes in the Tpm affinity for actin and only a slight increase in the thermal stability of the Tpm-actin complex were found. The similarity in the functional properties of D219V and E218L Tpm is not surprising, as both residues are probably involved in electrostatic and *h*-bonding interactions with K326 and K328 residues of actin (Figure 7 and Figure 8). In addition, the E218L substitution significantly increased the bending stiffness of Tpm estimated by MD simulation and the thermal stability of Tpm as the polar destabilizing non-canonical Tpm core residue E218 was substituted by a canonical hydrophobic one.

Another substitution of residue 219 of human cardiac Tpm, D219N, was registered in the ClinVar database (Variant ID 165571, last accessed 12 November 2022) as likely pathogenic (Class IV). It was found in a patient with a consistent phenotype (hypertrophic cardiomyopathy); however, the functional analysis was not provided, and the pathogenicity of this substitution is under consideration.

### 3.5. Limitation of the Work

The D219V mutation described here was heterozygous. Presumably, a mixture of the Tpm WT and Tpm D219V homodimers and the Tpm WT/D219V heterodimers were present in the cardiac muscle of the SCD victim. Our work was limited to studying various properties of the Tpm D219V homodimers and their comparison with those of the Tpm WT homodimers.

## 4. Materials and Methods

### 4.1. Genetic Investigation

Proband’s DNA was extracted from the venous blood during the molecular autopsy examination, and DNA samples of the parents were extracted from dry venous blood spots on DNA collection filters. Genomic DNA was extracted from biological samples using Thermo Scientific^TM^ Genomic DNA Purification kits (Thermo Scientific^TM^ is part of Thermo Fisher Scientific, Waltham, MA, USA) and following the manufacturer’s protocol. A genetic study for the proband was performed by target genes panel sequencing using the semiconductor sequencing platform PGM IonTorrent (Thermo Fisher Scientific, Waltham, MA, USA). Targeted genes mini panel “Hypertrophic cardiomyopathy” included *ACTC1, LDB3, MYBPC3, MYL2, MYL3, MYH7, TAZ, TPM1, TNNI3*, and *TNNT2*. Oligoprimers for this gene panel were designed using the Ion Ampliseq Designer® online tool (Thermo Fisher Scientific, Waltham, MA, USA). The reads were preprocessed using Torrent Suite Software 5.6.0 and the variant annotation web server Ion Reporter 5.12 (Thermo Fisher Scientific, Waltham, MA, USA). NGS sequencing reads were visualized using the Integrative Genomic Viewer (IGV) tool [41] with hg19 as a reference genome. All genetic findings detected by NGS in the proband were validated by capillary Sanger sequencing using a 3130 Genetic Analyzer (Thermo Fischer Scientific, Waltham, MA, USA).

Pathogenicity evaluation for the identified genetic variants was carried out according to the guidelines of the American College of Medical Genetics (2015) [21]. The detection of the genetic variant in the *TPM1* gene in both parents was performed by capillary Sanger sequencing. Pathogenicity of the c.656A>T (p.D219V) variant was assessed following ACMG (2015) criteria [21].

### 4.2. Protein Preparations

The Tpm species used in this work were recombinant proteins, which have an Ala-Ser N-terminal extension to imitate naturally occurring N-terminal acetylation of native Tpm [42]. Human Tpm1.1 WT and Tpm1.1 D219V mutants were prepared in bacterial expression plasmid pMW172 by PCR-mediated site-directed mutagenesis using Q5 DNA Polymerase (NEB, New England Biolabs, Hitchin, UK). The following oligonucleotides were used for obtaining the construct carrying the D219V mutation:

5′-GCAGAAGGAAGTCAGATATGAGGAAGAGATC-3′ as a forward primer (mutant codon is underlined) and 5′-GAGTACTTCTCAGCCTGAGCCTCCAGTG-3′ as an adjacent primer. The PCR products were cloned and sequenced to verify the substitution. Protein expression and purification were performed as described previously [43,44]. The Tpm concentration was determined spectrophotometrically at 280 nm using an E^1%^ of 2.7 cm^−1^.

Myosin was extracted from the left atrium and ventricle of the pig by the standard method [45]. The content of the myosin heavy (MHC) and light (LC) chain isoforms was determined by SDS-PAGE [46]. Atrial myosin contained ~95% α-MHC, ~5% β-MHC, and atrial LCs; ventricular myosin contained 100% β-MHC and ventricular LCs (see Appendix A). Bovine cardiac actin and troponin were prepared by standard methods [47,48]. For the in vitro motility experiments, F-actin was labeled with TRITC-phalloidin (Sigma Chemical Co., St. Louis, MO, USA).

### 4.3. Differential Scanning Calorimetry (DSC)

DSC experiments were performed as described [49] on a MicroCal VP-Capillary differential scanning calorimeter (Malvern Instruments, Northampton, MA, USA) at a heating rate of 1 K/min in 30 mM Hepes-Na buffer, pH 7.3, containing 100 mM NaCl. The protein concentration was 2 mg/mL. All Tpm species were reduced before DSC experiments by heating at 60 °C for 20 min in the presence of 3 mM DTT. After such a procedure, all Tpm samples were in an entirely reduced state [26,49]. The thermal unfolding of both Tpm WT and Tpm D219V was fully reversible, thus allowing deconvolution analysis of the heat sorption curves, i.e., their decomposition into separate thermal transitions (calorimetric domains), which was performed by fitting the non-two-state model to the data as described earlier [26,49].

### 4.4. Viscosimetry

The viscosity measurements were performed on a falling ball microviscometer Anton Paar AMVn (Anton Paar USA Inc., Ashland, VA, USA), in a 0.5 mL capillary at 20 °C. The specific density of the Tpm solutions was measured with an Anton Paar DMA 4500 device (Anton Paar USA Inc., Ashland, VA, USA) and taken into account for accurate viscosity calculation. All measurements were performed at a Tpm concentration of 2 mg/mL in a 30 mM Hepes-Na buffer (pH 7.3) containing 100 mM NaCl and 4 mM DTT. The measurements for each Tpm sample were repeated three times, and the obtained values for Tpm viscosity over buffer viscosity were averaged. 

### 4.5. Cosedimentation of Tpm Species with F-Actin

The apparent affinity of Tpm species to actin was estimated using a co-sedimentation assay as described [27,28]. Three measurements were performed for each Tpm sample and then averaged.

### 4.6. Temperature Dependences of Light Scattering

Thermally-induced dissociation of Tpm complexes with F-actin stabilized by phalloidin was detected by changes in light scattering at 90°, as described earlier [27,28]. The experiments were performed at a wavelength of 350 nm on a Cary Eclipse fluorescence spectrophotometer (Varian Australia Pty Ltd, Mulgrave, VI, Australia) equipped with a temperature controller and thermoprobes. All measurements were performed at a constant heating rate of 1 °C/min. The dissociation curves were fitted by a Boltzmann sigmoidal decay function. The main parameter extracted from this analysis is *T_diss_*, i.e., the temperature at which a 50% decrease in light scattering occurs.

### 4.7. In Vitro Motility Assay

Experiments in the in vitro motility assay were performed as described [27]. In brief, 300 µg/mL myosin in AB buffer (25 mM KCl, 25 mM imidazole, 4 mM MgCl2, 1 mM EGTA, and 20 mM DTT, pH 7.5) containing 0.5 M KCl was loaded into the experimental flow cell. After 2 min, 0.5 μg/mL BSA was added for 1 min. Further 50 µg/mL of non-labeled F-actin in AB buffer with 2 mM ATP was added for 5 min to block nonfunctional myosin heads. To form regulated thin filaments, 10 nM TRITC-phalloidin labeled F-actin and 100 nM of Tpm and Tn were added for 5 min. Finally, the cell was washed with AB buffer, containing 0.5 mg/mL BSA, an oxygen scavenger system, 20 mM DTT, 2 mM ATP, 0.5% methylcellulose, 100 nM Tpm/Tn, and appropriate Ca^2+^/EGTA in proportions calculated with the Maxchelator program. The experiments were performed at 30 °C. The sliding velocity was analyzed with GMimPro software [50] as described previously [27].

To assess the cooperative effects of the D219V Tpm mutation on the actin-myosin interaction, the dependence of the sliding velocity of the F-actin-Tpm filament on the myosin concentration (c) was analyzed as described [29]. The surface density of myosin was varied by the infusion of different myosin concentrations in the flow cell. The experiments were repeated 3 times, and the movement of 50–100 filaments was analyzed in each experiment at each myosin concentration.

The dependence of the sliding velocity *V* of the regulated thin filament on the concentration of myosin added to the flow cell was fitted with the Hill equation [51]: *V* = *V*_max_ × *c^h^* × (*c*_50_*^h^* + *c^h^*)^−1^ where *V*_max_ is the maximal sliding velocity, *c* is myosin concentration, *c*_50_ is the concentration required to achieve half-maximal velocity, and *h* is the Hill cooperativity coefficient.

### 4.8. Molecular Dynamics

The MD simulation was performed using GROMACS v. 2019.3 [52]. The structure of the regulatory unit of the thin filament in the absence of calcium (PDB code 6KN7 [31]) was used as a starting model. The D219V substitution was introduced with the UCSF CHIMERA (University of California, San Francisco, CA, USA) [53]. The model structures were immersed in a 13.5 × 18 × 50 nm^3^ rectangular box filled with water molecules. Na^+^ and Cl^−^ ions were added to provide zero net charge and an ionic strength of 0.15 M. To mimic forces and torques, which act on the ends of Tpm molecules from Tpm of neighbor regulatory units at both ends, the position restraints in the plane perpendicular to the filament axis were applied to the Cα atoms of three boundary amino acids of all 8 Tpm α-helices in the structure (Figure 7). As no restraints were applied to actin or Tn, these restraints did not prevent a rotation of the Tpm-Tn strand with respect to the filament axis or the blocked-to-close transition of the regulatory unit. The energy minimization, the equilibrations in the NVT and NPT ensembles, and the 200-ns long MD runs were performed with the AMBER99SB-ILDN force field [54]. The snapshots of the MD trajectory were recorded every 50 ps and then analyzed. To estimate the effect of the D219V Tpm mutation on the movement of the Tpm-Tn strand around the axis of the actin filament, we analyzed the MD trajectories by removing the solid-body motion of actin. 

## 5. Conclusions

We identified a new de novo variant, c.656A>T (p.D219V), in the *TPM1* gene. The D219V mutation significantly affects the properties of Tpm, causing a substantial increase in Ca^2+^ sensitivity and incomplete relaxation at low Ca^2+^ concentrations of the contractile apparatus of cardiac muscle, which is most likely a cause of the weakening of the Tpm interaction with actin in the "closed" state. Such changes are characteristic of genetic HCM. Based on these findings, we conclude that the D219V Tpm mutation is HCM-associated and likely pathogenic. For the ultimate determination and characterization of its pathogenicity, further investigations, including those on transgenic animals, would be helpful.

## Figures and Tables

**Figure 1 ijms-24-00018-f001:**
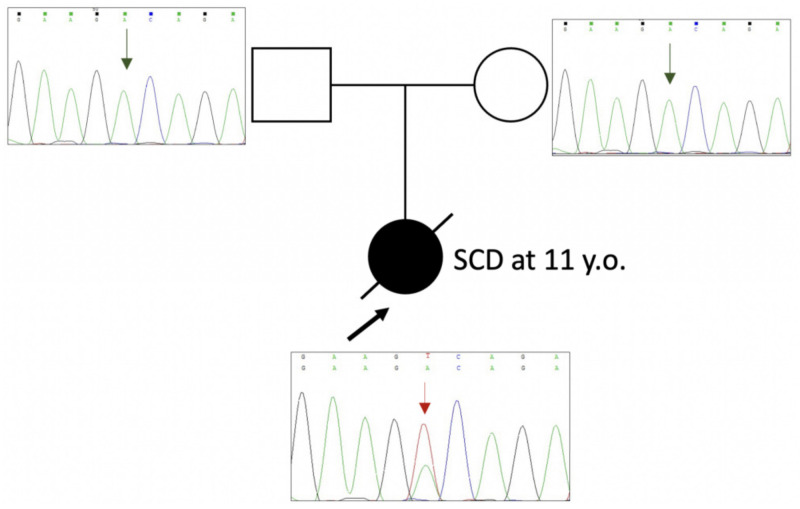
Pedigree of the family SD5. The affected family member is shown with closed and crossed-out symbols, and the unaffected parents with open symbols. Proband is marked by the black arrow. Fragments of the Sanger electropherograms of the *TPM1* gene are shown near the symbol of a family member. The letters in the upper row do not designate amino acids but nucleotides (A—adenine, green peaks; C—cytosine, blue peaks; G—guanine, black peaks; and T—thymine, red peaks). The heterozygous mutation c.656A>T (p.D219V) is marked by the red arrow.

**Figure 2 ijms-24-00018-f002:**
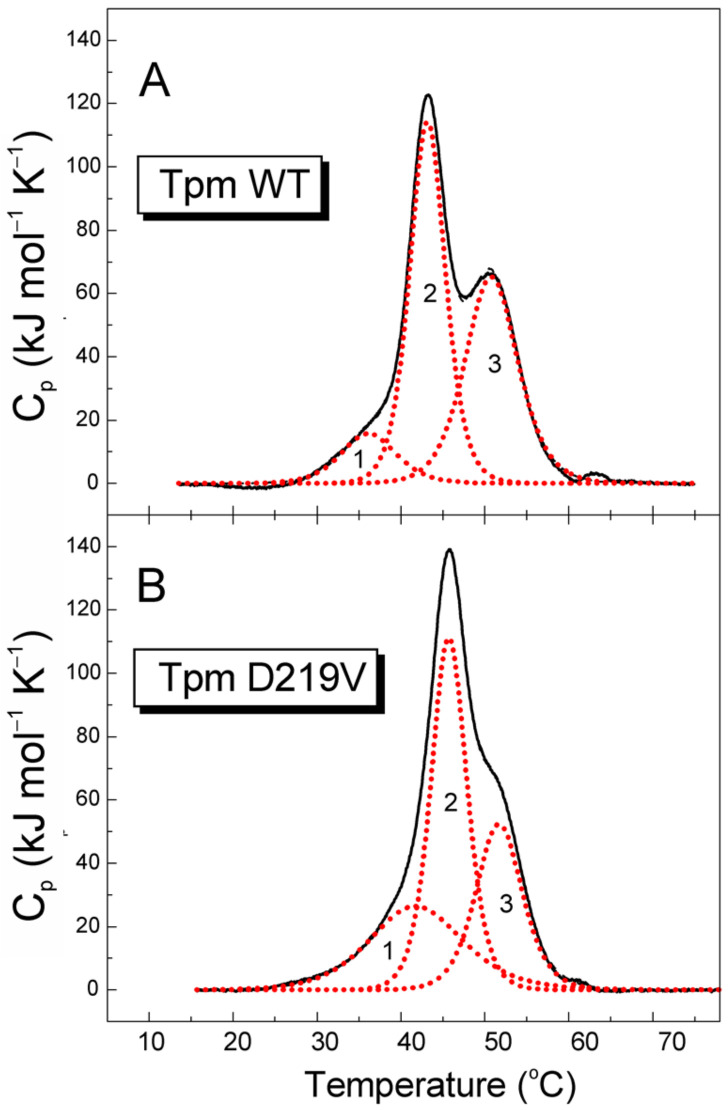
Temperature dependences of the excess heat capacity (*C_p_*) monitored by DSC and deconvolution analysis of the heat sorption curves for WT Tpm (**A**) and Tpm with mutation D219V (**B**). Solid lines represent the experimental curves after the subtraction of instrumental and chemical baselines, and dotted red lines represent the individual thermal transitions (calorimetric domains) obtained from fitting the non-two-state model [25] to the data.

**Figure 3 ijms-24-00018-f003:**
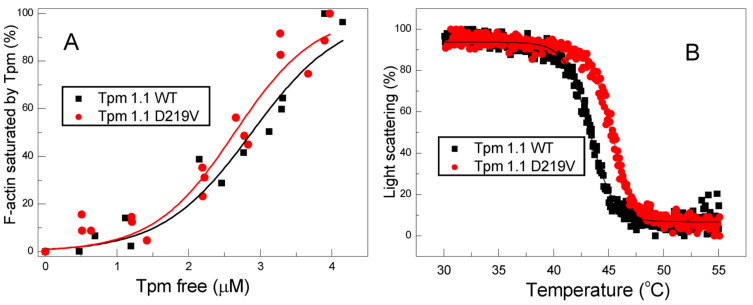
The effect of the D219V Tpm mutation on the Tpm affinity for F-actin was estimated by the co-sedimentation assay (**A**), and the thermal stability of the Tpm-actin complex was determined by light scattering (**B**).

**Figure 4 ijms-24-00018-f004:**
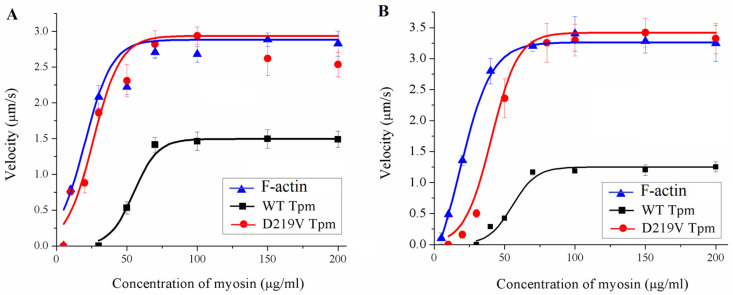
The effects of the D219V Tpm mutation on the dependence of the sliding velocity of the F-actin–Tpm filament on the concentration of ventricular (**A**) and atrial (**B**) myosin in the in vitro motility assay. Each data point represents the Mean ± SD from three experiments. The data are fitted with the Hill equation. The parameters of the Hill equation are presented in Table 2.

**Figure 5 ijms-24-00018-f005:**
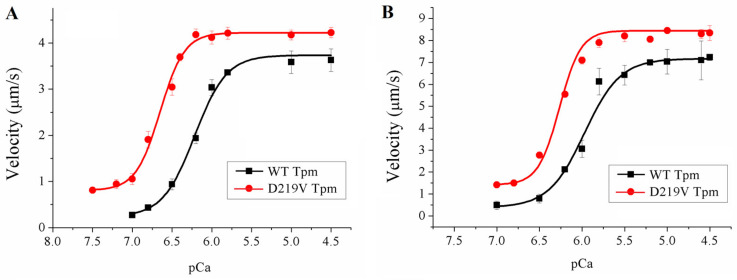
The effects of the D219V Tpm mutation on the Ca^2+^-dependent sliding velocity of regulated thin filaments moving over ventricular (**A**) and atrial (**B**) myosin in the in vitro motility assay. Each data point represents the mean ± SD from three experiments. The data are fitted with the Hill equation, and the parameters of the *p*Ca-velocity relationships are presented in Table 3.

**Figure 6 ijms-24-00018-f006:**
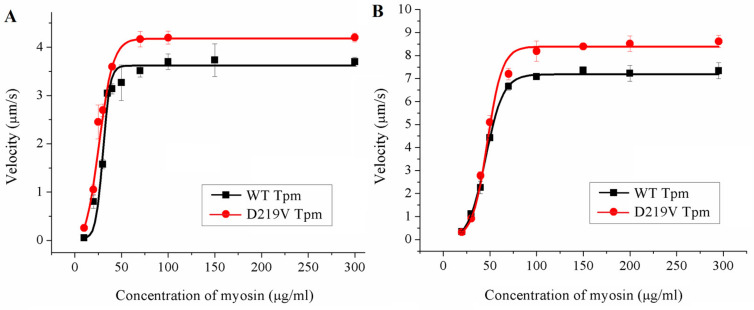
The dependence of the maximal sliding velocity of thin filaments with WT and D219V Tpm at saturated Ca^2+^ concentration on LV (**A**) and LA (**B**) myosin concentrations added to the flow cell. *C*_50_ for LV myosin with WT Tpm was 30.1 ± 0.9 µg/mL and 25.9 ± 3.1 µg/mL with D219V Tpm. The *C*_50_ for LA myosin with WT Tpm was 45.3 ± 1.5 µg/mL and 47.2 ± 1.1 µg/mL with D219V Tpm.

**Figure 7 ijms-24-00018-f007:**
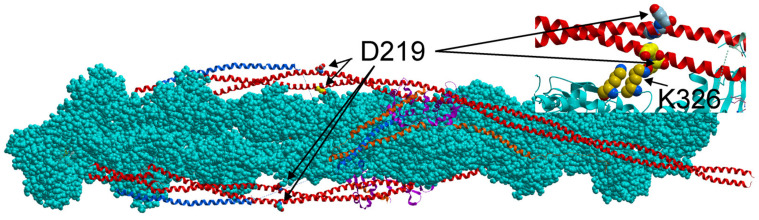
Location of the D219 Tpm residues in the structure of the regulatory unit of the thin filament of cardiac muscle ([31]; PDB code 6KN7). Actin atoms are shown by space-filling (CPK) representation (cyan); the Tpm molecules are shown by red ribbons; Tn-Cs are magenta ribbons; Tn-Is are orange ribbons; and Tn-Ts are blue ribbons. An *h*-bond between the D219 residue of Tpm and the K326 residue of actin is shown in the inset. Neighbor E218 residue of Tpm and the K328 residue of actin are also shown in CPK.

**Figure 8 ijms-24-00018-f008:**
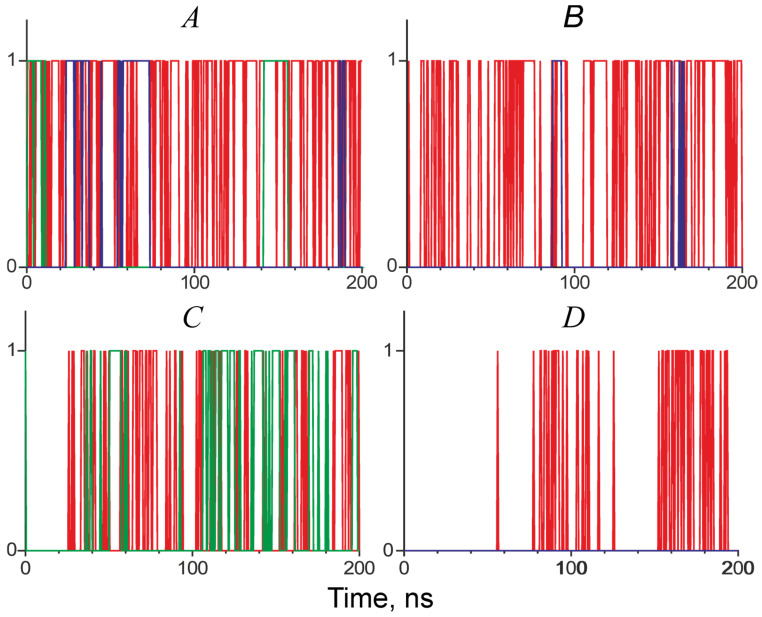
The time course of the existence of *h*-bonds between E218 and D (or V)219 Tpm residues and K326 and K328 residues of neighboring actin monomers during a 200 ns-long MD trajectory with WT Tpm (**A**,**C**) and D219V Tpm (**B**,**D**); vertical axis: one means at least one *h*-bond between the residues, zero otherwise; (**A**–**D**) correspond to two different Tpm strands. The *h*-bond between E218 of Tpm and K326 of actin is red; the *h*-bond between E218 of Tpm and K328 of actin is blue; and the *h*-bond between D (or V)219 Tpm residue and K326 of actin is green.

**Table 1 ijms-24-00018-t001:** Calorimetric parameters obtained from the DSC data for individual thermal transitions (calorimetric domains) of WT Tpm and Tpm with mutation D219V ^a^.

Tpm	*T*_m_^b^ (°C)	Δ*H*_cal_ (kJ mol^−1^)	Δ*H*_cal_ (% of Total)	Total Δ*H*_cal_ ^c^ (kJ mol^−1^)
**Tpm WT**				1330
Domain 1	36.2	140	11	
Domain 2	43.2	645	48	
Domain 3	50.8	545	41	
**D219V Tpm**				1405
Domain 1	41.9	390	28	
Domain 2	45.7	630	45	
Domain 3	51.7	385	27	

^a^ The parameters were extracted from the DSC curves shown in Figure 3. ^b^ The error of the given values of transition temperature (*T*_m_) did not exceed ±0.2 °C. ^c^ The relative error of the given values of calorimetric enthalpy, Δ*H*_cal_, did not exceed ±10%.

**Table 2 ijms-24-00018-t002:** The effects of the D219V Tpm mutation on the dependence of the sliding velocity of the F-actin–Tpm filament on the myosin concentration in the in vitro motility assay.

Myosin	Filament	*V*_max_ (µm/s)	*C*_50_ (µg/mL)
Ventricular myosin	F-actin alone	2.88 ± 0.01	20.1 ± 1.2
	WT Tpm	1.50 ± 0.01 *	54.9 ± 2.1 #
	D219V Tpm	2.93 ± 0.01	26.0 ± 1.0 *, #
Atrial myosin	F-actin alone	3.26 ± 0.03	18.0 ± 1.4
	WT Tpm	1.25 ± 0.01 *	55.4 ± 0.2 #
	D219V Tpm	3.32 ± 0.03	43.0 ± 0.1 *, #

*V*_max_, the sliding velocity of the F-actin–Tpm filament at saturated myosin concentration in the in vitro motility assay; and *C*_50_, the myosin concentration at which the sliding velocity is half-maximal. The * and # symbols denote statistically significant differences in characteristics between D219V Tpm and those of WT Tpm and F-actin, respectively (*p* < 0.05).

**Table 3 ijms-24-00018-t003:** The parameters of the *p*Ca-velocity relationship of the thin filament containing WT or D219V Tpm in the in vitro motility assay.

Myosin	Tpm	*V*_max_ (µm/s)	*V*_0_ (µm/s)	*p*Ca_50_
LV	WT Tpm	3.73 ± 0.13	0.24 ± 0.06	6.22 ± 0.03
	D219V Tpm	4.22 ± 0.05	0.81 ± 0.05 *	6.65 ± 0.02 *
LA	WT Tpm	7.17 ± 0.10	0.39 ± 0.17	5.97 ± 0.03
	D219V Tpm	8.45 ± 0.05 *	1.42 ± 0.01 *	6.27 ± 0.01 *

LV is the left ventricle and LA is the left atrium; *V*_0_ and *V*_max_ are sliding velocities at low and saturated Ca^2+^ concentrations, respectively; *p*Ca_50_ is the *p*Ca value at which the sliding velocity is half-maximal. The * symbol denotes statistically significant differences in characteristics of the D219V Tpm containing filaments from those with WT Tpm, *p* < 0.05 (*t*-test).

## Data Availability

All information related to the patient/family is completely anonymized and unidentifiable. The raw clinical data supporting this article cannot be placed in a public repository due to ethical reasons (personal data protection) but it will be available by co-author (E.V.Z., zhelene@mail.ru) upon reasonable request. The genetic data is registered in ClinVar (https://www.ncbi.nlm.nih.gov/clinvar/variation/1727219/?new_evidence=true). Accession numbers for this variant submission are VCV001727219.1, Variation ID: 1727219.

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
