# Peer review of "De Novo Asp219Val Mutation in Cardiac Tropomyosin Associated with Hypertrophic Cardiomyopathy"

_ijms, 2022, doi:10.3390/ijms24010018_

Round 1
Reviewer 2 Report
This manuscript by Tsaturyan and co-workers focuses on the characterization of recombinant Asp-219-Val tropomyosin Tpm-1.1, based on the recent finding that this mutation affects cardiac tropomyosin and appears to be associated with hypertrophic cardiomyopathy. Although this mutation does not appear to directly change the binding affinity between filamentous actin and the modified tropomyosin molecule, the calcium sensitivity during thin filament sliding was shown to be altered. These findings have potential biomedical implications for the interpretation of pathophysiological mechanisms that are involved in the disturbed interaction patterns of contractile proteins in hypertrophic cardiomyopathy.
The authors are encouraged to address the following issues:
Page 1 (Abstract): The term ‘TPM1’ should be in italics if it refers to the gene name, as in the rest of the manuscript.
Page 1 (line 345): Please rephrase the beginning of the sentence: ‘Numerous data show …’. This sentence relates to quoting references [6-15], so should probably better state ‘… Findings from a large number of studies have established that …’.
Page 2 (autopsy description): During the autopsy procedure, where there any cardiac tissue samples taken, which could be used for basic biochemical and cell biological studies? It would be interesting to see whether the mutation causes any abnormal biochemical or cell biological properties of cardiac tropomyosin, which can be easily examined using gel electrophoresis, immunoblotting, mass spectrometry and immunofluorescence microscopy. On line 80: ‘post mortem’ is usually written in italics.
Page 3 (Figure 3): The electrophoregramms of the TMP1 gene that are shown in the family pedigree tree should be properly labelled and ideally shown at a larger size so that details are better displayed.
Figures 3, 4 and 6; Protein preparations for experiments (see Methods section on Page 12): The authors should provide evidence for the purity of the extracted proteins myosin and actin. For myosin, data should be provided for the MHC composition. This can be provided as supplementary figures. Just referring to previous publications does not give any information on the actual probes used in this report. The description of myosin isolation is based on rabbit skeletal muscle. Was the method used in this study adapted for a different protocol?
Figure 7: The colour coding in this figure is of poor quality. The green arrows are difficult to differentiate from the greenish-blue colour of the underlying model of actin. Arrows could be given with a white edge or a different colour for a better presentation.
Page 11 (line 361): Please provide the name of the company that has provided the genomic DNA extraction kit.
Page 13 (Section 4.8.): The formatting of this section is incorrect and lacks a gap to the previous section.
Round 2
Reviewer 1 Report
The Authors responded adequately to all comments of this reviewer. The answers are an interesting discussion of the data, so it is somewhat surprising that no part of this discussion made its way into the manuscript itself. Because interested readers of the paper may have similar questions as this reviewer, discussing the various possibilities would not only dispel their doubts, but would also strengthen the interpretation of the results proposed by the Authors. Please, include the explanations described in your rebuttal letter in the Discussion section.
In addition, the Authors explained the problem of Tn denaturation, which hampers the analyses of thermal stability of the thin filament by light scattering. Since the Authors already have unpublished results showing that the TnT1 fragment increases the thermal stability of the F-actin-Tpm complex, I wonder what prevents them from repeating the experiment in the presence of the D219V Tpm mutant and TnT1 to actually see if the effect of the mutation is mitigated by TnT1? Such experimental data would provide a solid basis for interpreting the results. After all, in the heart, the thin filament consists of F-actin, Tpm and Tn complex.
Round 3
Reviewer 1 Report
The authors addressed my comments properly. The discussion of the revised manuscript shows effects of the mutation in a broader perspective.
English should be carefully revised,
e.g.: line 329 is "it is worse", should be "it is worth"
Author Response
English should be carefully revised, e.g.: line 329 is "it is worse" should be "it is worth"
This annoying typo on line 329 is corrected as suggested. We also did our best to edit English of our revised manuscript and to avoid typos.